# The Potential microRNA Prognostic Signature in HNSCCs: A Systematic Review

**DOI:** 10.3390/ncrna9050054

**Published:** 2023-09-14

**Authors:** Mario Dioguardi, Francesca Spirito, Giovanna Iacovelli, Diego Sovereto, Enrica Laneve, Luigi Laino, Giorgia Apollonia Caloro, Ari Qadir Nabi, Andrea Ballini, Lorenzo Lo Muzio, Giuseppe Troiano

**Affiliations:** 1Department of Clinical and Experimental Medicine, University of Foggia, Via Rovelli 50, 71122 Foggia, Italy; spirito.francesca97@gmail.com (F.S.); giovanna_iacovelli.559712@unifg.it (G.I.); diego_sovereto.546709@unifg.it (D.S.); enrica.laneve@unifg.it (E.L.); andrea.ballini@unifg.it (A.B.); lorenzo.lomuzio@unifg.it (L.L.M.); giuseppe.troiano@unifg.it (G.T.); 2Multidisciplinary Department of Medical-Surgical and Odontostomatological Specialties, University of Campania “Luigi Vanvitelli”, 80121 Naples, Italy; luigi.laino@unicampania.it; 3Unità Operativa Nefrologia e Dialisi, Presidio Ospedaliero Scorrano, ASL (Azienda Sanitaria Locale) Lecce, Via Giuseppina Delli Ponti, 73020 Scorrano, Italy; giorgiacaloro1983@hotmail.it; 4Biology Department, Salahaddin University-Erbil, Erbil 44001, Kurdistan, Iraq; ari.nabi@su.edu.krd

**Keywords:** microRNA, risk factor, HNSCC, oral cancer

## Abstract

Head and neck squamous cell carcinomas (HNSCCs) are often diagnosed at advanced stages, incurring significant high mortality and morbidity. Several microRNAs (miRs) have been identified as pivotal players in the onset and advancement of HNSCCs, operating as either oncogenes or tumor suppressors. Distinctive miR patterns identified in tumor samples, as well as in serum, plasma, or saliva, from patients have significant clinical potential for use in the diagnosis and prognosis of HNSCCs and as potential therapeutic targets. The aim of this study was to identify previous systematic reviews with meta-analysis data and clinical trials that showed the most promising miRs in HNSCCs, enclosing them into a biomolecular signature to test the prognostic value on a cohort of HNSCC patients according to The Cancer Genome Atlas (TCGA). Three electronic databases (PubMed, Scopus, and Science Direct) and one registry (the Cochrane Library) were investigated, and a combination of keywords such as “signature microRNA OR miR” AND “HNSCC OR LSCC OR OSCC OR oral cancer” were searched. In total, 15 systematic literature reviews and 76 prognostic clinical reports were identified for the study design and inclusion process. All survival index data were extracted, and the three miRs (miR-21, miR-155, and miR-375) most investigated and presenting the largest number of patients included in the studies were selected in a molecular biosignature. The difference between high and low tissue expression levels of miR-21, miR-155, and miR-375 for OS had an HR = 1.28, with 95% CI: [0.95, 1.72]. In conclusion, the current evidence suggests that miRNAs have potential prognostic value to serve as screening tools for clinical practice in HNSCC follow-up and treatment. Further large-scale cohort studies focusing on these miRNAs are recommended to verify the clinical utility of these markers individually and/or in combination.

## 1. Introduction

Among the main tumors of the head and neck region, oral squamous cell carcinomas (OSCCs) represent the sixth malignant tumor in global incidence, with about 700,000 thousand new cases each year [1].

The risk factors most associated with the onset of head and neck squamous cell carcinoma (HNSCC) are alcohol and the consumption of smoked or chewed tobacco, and for laryngeal squamous cell carcinoma (LSCC), positivity to HPV subtypes 16 and 18 was considered a risk factor but with a favorable prognosis [2].

Survival at 5 years after diagnosis remains very low, as only one of two patients survives, and surgical resective therapy can be very debilitating, with a worsening of the quality of life, difficulty in swallowing and speech, and in general due to a perceived deterioration in the relationship with other people [3].

The identification of survival prognostic biomarkers remains a very open topic: In fact, the ability to predict a disease by estimating the clinical trend and survival time remains one of the diagnostic and prognostic objectives to be achieved. In recent decades, several prognostic biomarkers have been investigated in an attempt to create a predictive survival biomolecular signature.

Among the widely studied prognostic and diagnostic biomarkers associated with head and neck cancers, we have the non-coding sequences of RNA messenger (mRNA), and among these, microRNAs (miRNA/miRs) [4]. The latter group is a class of mature, non-coding, single-stranded RNAs with 21–23 nucleotides, which were proposed as promising biomarkers for patients with cancer diagnosis and follow-up [3,5].

Some previous systematic literature reviews have tried to identify individual miRs, aggregating the prognostic survival data of multiple studies, obtaining promising results only in some cases for HNSCCs, such as in the cases of miR-31 [6], miR-21 [7], miR-155 [8], and miR-195 [9].

Other studies tried to identify a biomolecular signature by aggregating miRNAs in HNSCC tissue expression, exploring their use as potential biomarkers for cancer detection and/or prognosis [10,11].

This systematic review aims to identify retrospective and prospective clinical studies investigating the prognostic value of miR expression in HNSCC patients, as well as including data from previous systematic reviews with meta-analyses. From these studies, we selected the most promising miRs, inserting them into a biomolecular signature to test the prognostic value on a cohort of HNSCC patients according to the “The Cancer Genome Atlas” (TCGA) [12].

## 2. Results

### 2.1. Study Selection

The following research question guided the selection of the studies: Are there biomolecular signatures consisting of non-coding mRNA sequences (especially miRs) in the scientific literature, whose differential expression in HNSCC tumor tissues was indicative of a different prognosis in patient survival?

The research phase was carried out by consulting and extracting the bibliographic references on three databases, SCOPUS (2455), Science Direct (1367), PubMed (2505), and on a Cochrane Library register (5), providing a total of 6332 articles.

Filters were applied on PubMed and Scopus to selectively include literature reviews and meta-analyses, together with clinical studies. Subsequently, the bibliographic references of Scopus and PubMed were reported on EndNote X8, and the duplicates were removed, while further overlapping of the references were manually removed. The articles obtained were selected by reading the abstract and the title; this phase was also performed for Science Direct and the Cochrane Library, and the articles selected from these two sources were added to those chosen from PubMed and Scopus, and thus 117 potentially eligible records were obtained.

A further search of the gray literature (Google Scholar and Open Gray) and previous systematic reviews did not identify additional manuscripts for inclusion in the present systematic review (Figure 1). Records were independently screened by two authors (M.D. and A.B.), while dubious situations were addressed at the end of the selection by involving a third author (F.S.) to resolve potential conflicts.

The last update of the literature search was conducted on 13 August 2023.

In total, 76 clinical studies and 15 systematic reviews were included at the end of the inclusion process. We designed our strategy to be optimized for a sensitive and broad search, and the results of this selection are reported in a flowchart (Figure 1).

### 2.2. Data Characteristics: Systematic Review

The systematic reviews included were the following: Dioguardi et al., 2023 [9]; Dioguardi et al., 2022 [13]; Dioguardi et al., 2022 [14]; Dioguardi et al., 2022 [8]; Dioguardi et. al, 2022 [6]; Dioguardi et al., 2022 [7]; Irimie-Aghiorghiese et al., 2019 [15]; Lubov et al., 2017 [16]; Xie and Wu, 2017 [17]; Wang et al., 2019 [18]; Li et al., 2019 [19]; Qiu et al., 2021 [20]; Huang et al., 2021 [21]; Jamali et al., 2015 [22]; and Troiano et al., 2018 [23]. The selected studies reported the following resulting data: overall survival (OS); disease-free survival (DFS); recurrence-free survival (RFS); cancer-specific survival (CSS); progression-free survival (PFS), and relative risk (RR).

On average, the selected reviews included many studies (≅9.1), with a range from 1 to 36, and the number of included patients ranged from 80 to 1200. Although the systematic review included HNSCCs, two reviews involved only OSCCs, and one study only covered LSCCs. The most reviewed prognostic index was the HR of OS (across different miR tissue expression levels), with thirteen reviews, followed by DFS (six studies), RFS (three studies), CSS (two studies), and PFS (one study). Only one review evaluated the RR of OS.

The miRs subjected to meta-analyses in the 15 systematic reviews were 64 (miR-205, miR-429, miR-21, miR-331-3p, miR-200a, miR-19a, miR-151a, miR-17, miR-18b, miR-324, miR-96, miR-29c, miR-200b, miR-375, miRNA-204, miR−200c, miR-130a, miR-15b, miR-203, miR-195, miR-300, miR-146, miR-155, miR-16-2, miR-10a, miR-100, miR-101, miR-34c, miR-125, miR-149, miR-145, miR-181a, let-7a, miR-494, miR-720, miR-675, miR-137, miR-31, miR-9, miR-424, miR-23a, miR-196b, let-7g, miR-210, miR-20a, miR-126, miR-205, miR-134a, let-7b, miR-153, miR-18a, miR-17a, miR-451, miR-193b, miR-455, miR-372, miR-373, miR29b, miR-1246, miR-196a, miR-181, miR-32, miR-16, and miR-125b).

The microRNAs most reviewed and included in the signatures were miR-21 (in eight revisions including four as a single miR), miR-155 (four times, three of which were within a signature with multiple miRs), and miR-375 (two within a signature). In particular, miR-21 was the most studied and (taken individually) presented an HR of OS ranging from 1.29 [7] to 1.81 [16].

All extracted data are shown in Table 1.

### 2.3. Data Characteristics: Clinical study

The clinical studies included in the review were 76: Jung et al., 2012 [24]; Kawakita et al., 2014 [25]; Hedbäck et al., 2014 [26]; Yu et al., 2017 [27]; Supic et al., 2018 [28]; Jakob et al., 2019 [29]; Li et al., 2013 [30]; Zheng et al., 2016 [31]; Li et al., 2009 [32]; Ganci et al., 2016 [33]; Wang et al., 2018 [34]; Qiang et al., 2019 [35]; Tu et al., 2021 [36]; Hess et al., 2017 [37]; Zhao et al., 2018 [38]; Baba et al., 2016 [39]; Shi et al., 2015 [40]; Kim et al., 2018 [41]; Bersani et al., 2018 [42]; Wu et al., 2020 [43]; Shuang et al., 2017 [44]; Ding and Qi, 2019 [45]; Jia et al., 2013 [46]; Qin et al., 2019 [47]; Liu et al., 2013 [48]; Maruyama et al., 2018 [49]; Zhao et al., 2018 [50]; Luo et al., 2019 [51]; Ahn et al., 2017 [52]; Hudcova et al., 2016 [53]; Kang et al., 2021 [54]; Bonnin et al., 2016 [55]; Ganci et al., 2013 [56]; Harris et al., 2012 [57]; Ahmad et al., 2019 [58]; Rajthala et al., 2021 [59]; Song et al., 2020 [60]; Zhao et al., 2018 [61]; Li et al., 2013 [62]; de Jong et al., 2015 [63]; Fang et al., 2019 [64]; He et al., 2017 [65]; Re et al., 2015 [66]; Xu et al., 2016 [67]; Tian et al., 2014 [68]; Zhao et al., 2018 [69]; Guan et al., 2016 [70]; Avissar et al., 2009 [71]; Wu et al., 2014 [72]; Wu, Zhang et al., 2014 [73]; Zhang et al., 2015 [74]; Hu et al., 2015 [75]; Re et al., 2017 [76]; Shen et al., 2012 [77]; Maia et al., 2017 [78]; Ogawa et al., 2012 [79]; Pantazis et al., 2020 [80]; Childs et al., 2009 [81]; Ko et al., 2014 [82]; Arantes et al., 2017 [83]; Chang et al., 2013 [84]; Gee et al., 2010 [85]; Jia et al., 2014 [86]; Liao et al., 2013 [87]; Liu et al., 2013 [88]; Liu, Shen et al., 2013 [89]; Luo et al., 2014 [90]; Peng et al., 2014 [91]; Sasahira et al., 2012 [92]; Tu et al., 2015 [93]; Wu et al., 2014 [94]; Xu et al., 2013 [95]; Zhang et al., 2017 [96]; Jia et al., 2015 [97]; Hu et al., 2014 [98]; and Gu et al., 2018 [99].

The total number of included patients affected by HNSCCs was 6848, with 3295 cases definitely identified as OSCCs and at least 1493 presenting a localization to the tongue, while LSCC was present in 2179 patients.

The most used prognostic indices were OS in 51 studies, DFS in 27 studies, RFS in 12 studies, and CSS in 6 studies. For risk factors, only 13 studies investigated HPV positivity, and 31 studies investigated t-7d, Let-7g, miR-9, miR-15b, miR-17, miR-18a, miR-18b, miR-19, miR-19a, miR-20b, miR-20a, miR-21, miR-22, miR-23a, miR-26a, miR-29b, miR-29c, miR-31, miR-34c, miR-34a, miR-375, miR-331, miR-324, miR-296, miR-205, miR-203, miR-204, miR-210, miR-1246, miR-675, miR-451, miR-452, miR-429, miR-422a, miR-134, miR-126, miR-300, miR-372, miR-373, miR-218, miR-153, miR-155, miR-181a, miR-183, miR-200c, miR-200b, miR-200c, miR-200a, miR-96 miR-195, miR-196a, miR-196a2, miR-196b, miR-197, miR-198, miR-151a, miR-146a, miR-99a, miR-99b, miR-100, miR-101, miR-141, miR-143, miR-145, miR-149, miR-130b, and miR-139.

Among these, the most studied were miR-21 (19 studies), miR-155 (8 studies), and miR-375 (7 studies).

In miR-21, the HR of OS between high and low expression levels ranged from 5.31 95% CI: [1.39–20.38] [24] to 1.1302 95% CI: [0.34–3.757] [98], and in poor OS, it was upregulated, similar to miR-155, while in miR-375, the HR of OS between low and high expression levels ranged from 12.8 95% CI: [3.4–48.6] [57] to 1.32 95% CI: [0.76–2.27] [53], and in case of low survival, it was downregulated. All data related to the clinical studies, as well as the survival data extrapolated from the Kaplan–Meier survival curves, are extensively reported in Table 2 and Table 3.

Analyzing the studies and the systematic reviews performed on the prognostic biomarkers of survival, it becomes clearly evident that the miR that has been most investigated and provides the greatest number of data is miR-21, with 19 clinical studies, followed by miR-155 (8 studies) and miR-375 (7 studies). The other miRs have fewer studies with fewer patients included than miR-21 (1262 patients), miR-155 (706 patients), and miR-375 (572 patients).

For this reason, we decided to use a different cohort (TCGA), which includes about 512 patients, to verify whether a biosignature with a high expression of these three miRs in tumor tissues was correlated with low survival, and significant results were obtained for miR-21 and miR-155, while for miR-375, low survival was associated with low expression.

Using the extracted data shown in Table 1 and Table 2, the three main miRs investigated in the literature (miR-21, miR-155, and miR-375), whose altered expression was investigated in the prognosis of survival in HNSCC patients and included in molecular biosignatures, were then selected. The evaluation was performed through the Kaplan–Meier plotter database portal (https://kmplot.com/analysis/, accessed on 10 May 2023) [100], and HR data were extracted.

The difference between high and low tissue expression levels of the miRs taken into consideration presents an HR of OS = 1.28 95% CI: [0.95, 1.72], log-rank *p* = 0.1. Moreover, the Kaplan–Meier survival curve generated using the portal is depicted in Figure 2, and the considered follow-up period was 60 months. The median survival in the cohort of patients with low expression was 58.73, while that in patients with high expression was 46.47.

The cut-off value between high and low miR expression levels was automatically generated through the portal (Figure 3), and the cut-off values and the related *p*-values are present in the the Appendix A.

The portal to generate and display the Kaplan–Meier plot is used to establish a cut-off value and assign samples to one of the two cohorts, using the best available cut-off value.

To find the best cut-off, the process is repeated using the values of the input variables from the lowest quartile to the upper quartile, and the Cox regression for each setting is calculated [101].

The most significant cut-off value was employed as the optimal threshold to segregate the input data into two groups. Subsequently, the system presents a straightforward visual representation of this analysis, displaying the *p*-values obtained concerning the selected cut-off values.

In cases where the generated cut-off values were ambiguous (e.g., multiple cut-off values resulted in very low *p*-values), the value corresponding to the highest hazard ratio was selected.

The calculation of multiple cut-off values led to the generation of multiple assumptions.

Hence, in this setup, the FDR was automatically computed using the Benjamini–Hochberg method to correct for multiple hypothesis testing [102].

Spearman’s correlation and Pearson’s correlation between the expression values of the different miRs investigated (Table 4 and Table 5) were also calculated.

All reported data can be reproduced via the Kaplan–Meier plotter portal [101].

Furthermore, additional tests were performed on miR-21, and for the main downregulated miRs described in the literature, the extrapolated data are described in Figure 4.

Furthermore, the main results related to the resistance to chemotherapy and radiotherapy administered to patients, in support of resective surgical treatment and in relation to the altered expression of microRNAs, were extracted from the studies, as shown in Table 6.

### 2.4. Risk of Bias

The risk of bias for systematic reviews was determined using the ROBIS tool, and for each factor, it was evaluated as “low”, “high”, or “unclear”. The three phases of the evaluation process were as follows: Phase 1: the evaluation of the relevance of the research question (PICO); Phase 2: the identification of critical points of the review process; and Phase 3: the evaluation of the overall risk of bias of the review. All data related to the risk of bias are reported in Table 7.

The main critical issues related to the individual revisions are as follows:➢Irimie-Aghiorghiese et al., 2019 [15]: Study eligibility criteria (?): The protocol number with which the systematic review was registered was not reported. Identification and selection of studies (?): The selection was performed only on two databases (PubMed and Embase), and the number of authors who conducted the research was not specified, nor were the start or end dates in which the review was conducted.➢Lubov et al., 2017 [16]: Study eligibility criteria (?): The protocol number with which the systematic review was registered was not reported. Identification and selection of studies (?): The number of authors who selected the articles and the start or end dates of the review were not reported. Data collection and study appraisal (?): The number of authors who performed the data extraction and the methods of data extraction were not stated. The manuscript is both a systematic review and a retrospective study of 100 patients.➢Xie and Wu, 2017 [17]: Study eligibility criteria (?): The protocol number with which the systematic review was registered was not reported.➢Wang et al., 2019 [18]: Study eligibility criteria (?): The protocol number with which the systematic review was registered was not reported.➢Li et al., 2019 [19]: Study eligibility criteria (?): The protocol number with which the systematic review was registered was not reported. Identification and selection of studies (?): The start or end dates of the review were not specified. Data collection and study appraisal (?): The risk of bias was not formally assessed using an appropriate scale or tool. A bioinformatic analysis was also performed.➢Huang et al., 2021 [21]: Study eligibility criteria (?): The protocol number with which the systematic review was registered was not reported. Identification and selection of studies (?): The start and end dates of the review were not specified.➢Jamali et al., 2015 [22]: Study eligibility criteria (?): The protocol number with which the systematic review was registered was not reported.➢Troiano et al., 2018 [23]: Study eligibility criteria (?): The protocol number with which the systematic review was registered was not reported.➢Dioguardi et al., 2023 [9]: Synthesis and findings (?): The obtained results were excessively emphasized in the conclusions.

The risk of bias for prognostic studies was assessed using the parameters derived from REMARK. According to the REMARK guidelines, a score ranging from 0 to 3 was considered for each factor (Table 8).

## 3. Materials and Methods

### 3.1. Protocol

The planning of the systematic review was implemented following the guidelines described in the Cochrane Handbook for Systematic Reviews of Interventions. The drafting of the review manuscript followed the recommendations of PRISMA (Preferred Reporting Items for Systematic Reviews and Meta-Analysis) [103], and the protocol was registered on PROSPERO before carrying out the selection of articles and was registered with the registration number CRD42023400856.

### 3.2. Eligibility Criteria

The search was directed towards the identification of retrospective or prospective clinical studies and bibliographic sources that reported systematic reviews of the literature regarding the role of non-coding RNAs and in particular of miRs, reporting prognostic data of survival in patients with HNSCCs associated with altered expression of a single miR or a signature of miR.

The exclusion criteria were the exclusion of all clinical trials and systematic reviews reporting no data on the use or detection of a molecular biosignature consisting of miRs in HNSCCs, all literature reviews (considered as bibliographic sources only), and studies that did not have an abstract in English.

Thus, the reporting data of all clinical trials and meta-analyses on a biomolecular signature consisting of miRs that is prognostic of survival in HNSCCs were considered potentially eligible.

The systematic review involved two reviewers (M.D. and A.B.) and followed the following stages:Choice of reviewers (M.D. and A.B.) and a third reviewer (F.S.) as a supervisor in case of conflict regarding the studies to be included, choice of outcomes to identify, choice of databases and k words used, choice of criteria of admissibility, choice of data to be extracted and methods of synthesis and registration of the protocol on PROSPERO;Identification of records and selection of studies through databases with the removal of duplicates performed manually or by software (EndNote 8.0), performed independently and subsequently comparison of selected studies and decision of studies to be included;Independently performed table data extraction and subsequent data comparison to minimize the risk of error in reporting information.

### 3.3. Sources of Information, Research, and Selection

The keywords used were microRNA AND HNSCC, LSCC AND MicroRna, OSCC AND MicroRna, and signature microRNA AND HNSCC.

The search was conducted on 3 databases, namely Science Direct, SCOPUS, and PubMed, and one registry, the Cochrane Library. Additionally, Google Scholar (keywords microRNA), gray literature sources such as Open Gray (keywords microRNA), and references from previous systematic reviews on miRs and HNSCCs were searched.

Particularly, the following are all the keywords used in the PubMed search:

Search: (signature microRNA OR miR) AND (HNSCC OR LSCC OR OSCC OR oral cancer) Sort by: Most Recent.

(((“protein domains”[MeSH Terms] OR (“protein”[All Fields] AND “domains”[All Fields]) OR “protein domains”[All Fields] OR “signature”[All Fields] OR “signatures”[All Fields]) AND (“microrna s”[All Fields] OR “micrornas”[MeSH Terms] OR “micrornas”[All Fields] OR “microrna”[All Fields])) OR (“med int rev”[Journal] OR “manag int rev”[Journal] OR “mir”[All Fields])) AND (“hnsccs”[All Fields] OR “squamous cell carcinoma of head and neck”[MeSH Terms] OR (“squamous”[All Fields] AND “cell”[All Fields] AND “carcinoma”[All Fields] AND “head”[All Fields] AND “neck”[All Fields]) OR “squamous cell carcinoma of head and neck”[All Fields] OR “hnscc”[All Fields] OR “LSCC”[All Fields] OR “OSCC”[All Fields] OR (“mouth neoplasms”[MeSH Terms] OR (“mouth”[All Fields] AND “neoplasms”[All Fields]) OR “mouth neoplasms”[All Fields] OR (“oral”[All Fields] AND “cancer”[All Fields]) OR “oral cancer”[All Fields])).

#### Translations

Signature: “protein domains”[MeSH Terms] OR (“protein”[All Fields] AND “domains”[All Fields]) OR “protein domains”[All Fields] OR “signature”[All Fields] OR “signatures”[All Fields].

microRNA: “microrna’s”[All Fields] OR “micrornas”[MeSH Terms] OR “micrornas”[All Fields] OR “microrna”[All Fields].

miR: “Med Int Rev”[Journal:__jid0017632] OR “Manag Int Rev”[Journal:__jid101593556] OR “mir”[All Fields].

HNSCC: “hnsccs”[All Fields] OR “squamous cell carcinoma of head and neck”[MeSH Terms] OR (“squamous”[All Fields] AND “cell”[All Fields] AND “carcinoma”[All Fields] AND “head”[All Fields] AND “neck”[All Fields]) OR “squamous cell carcinoma of head and neck”[All Fields] OR “hnscc”[All Fields].

Oral cancer: “mouth neoplasms”[MeSH Terms] OR (“mouth”[All Fields] AND “neoplasms”[All Fields]) OR “mouth neoplasms”[All Fields] OR (“oral”[All Fields] AND “cancer”[All Fields]) OR “oral cancer”[All Fields].

The literature search was completed on 20 February 2023.

The data to be extracted included the first author of the study, the publication date, the country in which the research was conducted, the type of squamous cell carcinoma, the number of patients involved in the study, the clinical characteristics of the patients and tumors included in the studies, data on the positivity to the HPV virus and exposure to risk factors such as smoking and alcohol, as well as clinical data on the staging of patients included in the studies and on the average or maximum follow-up, risk of bias tools, the studied miRs, the value or type of risk rate (RR) or hazard rate (HR) for various prognostic survival indices: overall survival (OS), disease-free survival (DFS), progression-free survival (PFS), relapse-free survival (RFS), and cancer-specific survival (CSS).

### 3.4. Risk of Bias, Bioinformatic Analysis

Furthermore, the data of a cohort of patients with HNSCCs (N ≈ 512) extracted from the Cancer Genome Atlas (TCGA) database were analyzed to obtain the HR values and associate the prognosis indices with the expression of the signature of miRs created and selected by the authors.

The risk of bias in the individual systematic reviews was assessed by two authors (M.D. and A.B.). The ROBIS (Risk of Bias in Systematic Reviews) was used as an assessment tool specifically developed to assess the risk of bias in systematic reviews. Studies with a high risk of bias were excluded from the review [104].

Clinical studies with the risk of bias were evaluated by two authors (M.D. and A.B.), and the tool used for the assessment of the parameters was derived from the reporting recommendations for prognostic studies of markers (REMARK). Studies with a high risk of bias were excluded from the analysis [105].

## 4. Discussion

In the last 40 years, a considerable number of new cancer biomarkers have been identified, but only a few have managed to be effectively used in clinical practice.

Many biomarkers pass validation very well, with concordant and reproducible results across trials; nevertheless, these biomarkers lack the capacity to decisively contribute to patient care, except to provide some additional information on prognosis. Therefore, they are considered by clinicians to be not fundamental in the therapeutic choice.

Physicians often show a tendency to overtreat specific patients, rather than relying on prognostic biomarkers that offer less than precise predictions. Using these flawed prognostic biomarkers could result in fewer patients receiving overly aggressive treatments (true positives) but, at the same time, could also increase the chance of not treating some patients who may actually benefit from therapy (false negatives). Therefore, from a clinical point of view, with the exclusion of fraudulent situations and sensationalist discoveries, the prognostic biomarkers detected are not so promising, and their failure is due to their inadequate performance in clinical practice.

Hence, in the process of transitioning a promising biomarker from Phase 1 studies to clinical implementation, meticulous consideration should be given to the study’s design, aiming to mitigate bias in the utilization of sensitive, specific, and precise analytical methodologies. This involves the careful selection of suitable samples, both in terms of quantity and quality, as well as appropriate patient subgroups for the purpose of validation. Furthermore, it is imperative to apply statistically robust and rigorous methods to prevent the occurrence of data overfitting.

The data present in the literature demonstrate how miRs are stable, and the results deriving from the studies are consistent and reproducible, making miRs potential promising biomarkers for diagnosis and prognosis [106].

Prognostic biomarkers including miRs could have a significant impact in helping clinicians improve the quality of life and health conditions of HNSCC patients, providing useful information for oncologists in terms of the most appropriate therapeutic choice, according to life expectancy and neoplasm aggressiveness [107].

Knowledge of the prognostic potential of biosignatures could be useful for clinicians after the diagnosis of HNSCCs to define prognosis by formulating predictive models of individualized prognostic risk. Bringing this model back into clinical practice in patients with HNSCCs who have unfavorable prognostic biosignatures (with low RFS or OS), a more or less aggressive therapy or surgical treatment could be recommended, with a tailored therapeutic approach in the context of personalized medicine [108].

The discovery of the miRs’ prognostic value presents critical insights with potential biases that must be taken into consideration before, during, and after the execution of retrospective studies or clinical trials, but also during the data meta-analysis. The choice of variables can significantly affect the results as well as the overall validity of the analysis.

Factors such as sample size, the heterogeneity of patient populations, virus positivity (HPV and EBV), and the choice of statistical analysis can affect the results.

In the context of HNSCCs, taking, for instance, HPV positivity as a variable, the role of papillomavirus as a risk factor in a subset of head and neck cancers [109], mainly oropharyngeal and laryngeal cancers, has been established, with different epidemiological, clinical, and molecular characteristics compared with HNSCCs, starting with HPV positivity, which was associated with distinctly different and more favorable prognostic survival values [110].

Therefore, the inclusion or exclusion of some clinical variables (smoking, alcohol, age, and gender) may alter the results of prognostic values for the associations observed between miR signatures, including the related survival results [111].

The meta-analysis size of the sample can also be addressed by performing a trial sequential analysis (TSA) to verify the power of the results as a function of the sample, with an effect achieved in terms of RR [112].

In addition, some laboratory study phases can be biased, making the detection of biomolecular signatures in biological samples difficult. In fact, possible biases can be identified in the sample selection (e.g., fresh tissue, fixed tissue, and biological fluids), RNA extraction, and sample quality control. In addition, miR profiling can also be affected by variability in the technical platform (instruments and software), which is an important source of bias that affects not least the data analysis [113].

In addition, the results of tissue miR expression seem to be influenced by the tissue preservation technique (frozen or in formalin); in fact, to reduce the heterogeneity of the data, it is recommended to aggregate data during a meta-analysis by conducting a subgroup analysis also based on the category of tissue preservation [114].

The difficulty in determining the prognostic value of miRs is due to the complexity of biological systems and the multiple roles of miRs in the regulation of gene expression. It is important to remember that microRNA expression patterns can vary between different cancer types, and even within subtypes of the same cancer, making it difficult to establish universal prognostic markers [113].

Furthermore, many miR biosignatures are currently being developed using algorithms and machine learning based on the search for associations between expression and disease outcomes. Therefore, causality is often not considered, and algorithms can generate signatures that are not biologically expressive, despite their statistical significance [115].

In this context, the execution of systematic reviews with the inclusion of Phase 2 prognostic studies can lead to improvement in these studies by better highlighting the most reliable and predictable results while not overlooking data without statistical significance or evidence (publication bias). The present systematic review aims to refine the design, execution, and reporting of Phase 2 studies [116,117] and provide useful knowledge in guiding Phase 3 clinical studies aimed toward finding a prognostic model [118].

Jamail et al. indicated that the over- or underexpression of some miRs was related to the survival of patients with HNSCCs, reporting that the elevated expressions of miR-21, miR-18a, miR-134a, miR-210, miR-181a, miR-19a, and miR-155 were associated with a reduction in the survival of patients with HNSCCs, while the decreased expression of miR-153, miR-200c, miR-363, miR-203, miR-17, miR-205, miR-Let-7d, Let-7g, miR-34a, miR-126a, miR-375, miR-491-p5, miR-218, miR-451, and miR-125b was associated with a poor survival prognosis [22].

These results agree with the findings of Huang et al. (2021) [21], who provide evidence in their review of LSCC suggesting that miRNA-100, miR155, miR-21, miR-34a, miR-195, and miR-let-7 are potential tumor biomarkers.

In light of the data reported in the medical literature, and from the preliminary research conducted in the field [6,7,8,9,13,14,119], we carried out our review after registering it on Prospero, which was written following the indications of PRISMA. A meta-analysis of the data was not carried out due to the excessive heterogeneity of data and histological subtypes of HNSCCs, and thus a TCGA analysis was instead used to test possible microRNA biosignatures that emerged from the data extraction and qualitative analysis of the studies.

In this systematic review, we identified 64 miRs from 15 systematic reviews, whose altered expression was correlated with prognostic indices. The miRs mainly investigated were miR-21, miR-155, and miR-375. HR values for OS in miR-21 ranged from 1.29 to 1.72, and these values were 1.59 for miR 375 and 1.40 for miR-155 (considering only the results of meta-analyses reporting HR values aggregated for individual miRs); the HR values for several miR panels ranged from 2.65 to 1.10 (Table 1).

By selecting the three miRs that, based on our research, were the most investigated in HNSCCs, and performing a survival analysis using these three miRs on the patient cohorts present in the TCGA, an HR of OS equal to 1.28 was found. From these preliminary data, it is evident that the existing results in the literature are still insufficient to clearly define a prognostic microRNA biosignature, and the retrospective statistical analyses performed using the TCGA in an attempt to further validate the findings do not fully achieve this purpose. In fact, by considering only miR-21, three meta-analyses report an aggregate HR value of about 1.7 as the difference between high and low expression levels, while using the TGCA, considering a follow-up period of 60 months, miR-21 presented an HR (high and low expression) equal to 1.27 95% CI: [0.95, 1.71] (Figure 4). Considering instead the HR data of miR-21, miR-155, and miR-375 using the TCGA and combining them in a single prognostic signature, the value of HR was 1.28 95% CI: [0.95, 1.72].

The performance of miRs is more or less superimposable if we consider the miRs that are reportedly downregulated in the literature during HNSCCs; for instance, Jamali et al. (2015) [22] and Wang et al. (2019) [18] revealed that hsa-miR-153 (-), hsa-miR-200c (-), hsa-miR-363 (-), hsa-miR-17 (-), hsa-miR-205 (-), hsa-Let-7d (-), hsa-Let -7g (-), hsa-miR-34a (-), hsa-miR-375 (-), hsa-miR-491 (-), hsa-miR-218 (-), hsa-miR-125b (-), and hsa-mir-375 (-) were downregulated, with an HR = 0.66 95% CI: [0.46–0.88] (Figure 4). Considering the HR between low and high expression levels, an HR of 1.51 was observed. These results are largely reproducible using the Kaplan–Meier portal except for subsequent updates of the latter.

## 5. Conclusions

In conclusion, we can state that although prognostic survival biomarkers have been identified that possess a discrete potential consisting of a miR signature, in the current state of knowledge for head and neck tumors, there are no studies that fully validate the results. Nevertheless, it is crucial to emphasize that additional validation is necessary before we can definitively establish their practicality. While some miRNA studies have revealed noteworthy findings related to their influence on patient survival, the limited number of studies that have been agregaded to derive these results diminishes their relevance in clinical contexts. Hence, there is a clear need for more extensive and long-term patient studies that specifically investigate these miRs.

## Figures and Tables

**Figure 1 ncrna-09-00054-f001:**
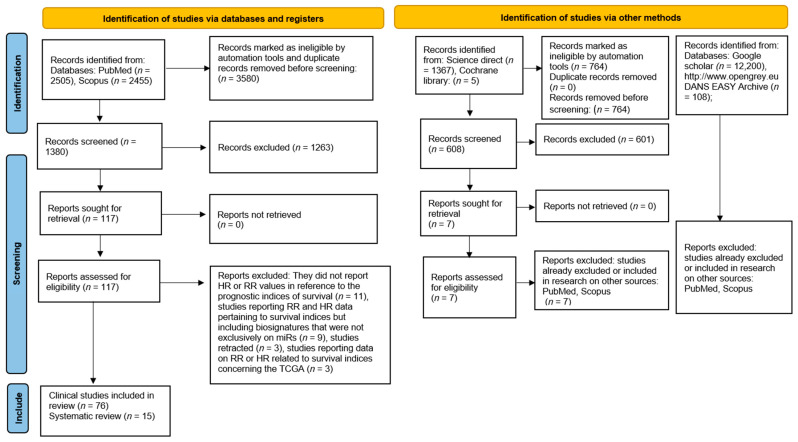
Flowchart describing the mechanisms of screening miR studies and including several databases and records.

**Figure 2 ncrna-09-00054-f002:**
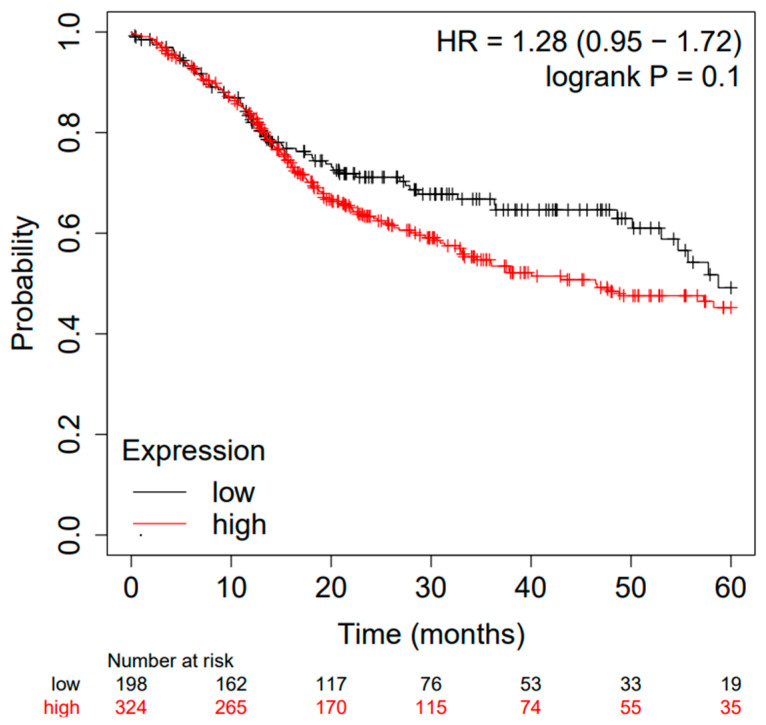
Kaplan–Meier curve based on miR-21, miR-155, and miR-375 expression levels for overall survival (OS) of patients with HNSCC (TCGA cohort); false discovery rate (FDR): 100%. Kaplan–Meier curves created from a public database and Kaplan–Meier plotter web application (http://kmplot.com/analysis/, accessed on 10 May 2023).

**Figure 3 ncrna-09-00054-f003:**
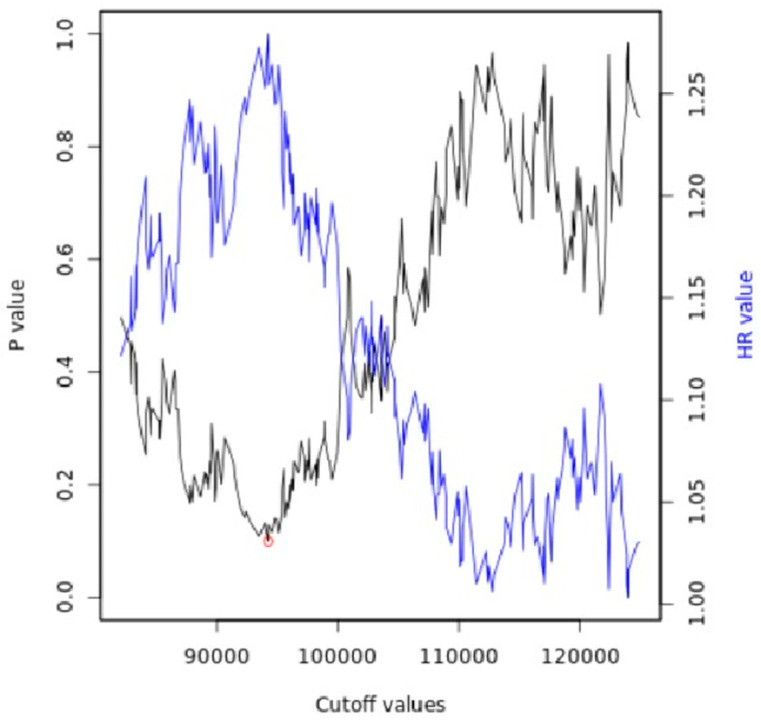
Automatically generated cut-off plot using the Kaplan–Meier plotter web application, http://kmplot.com/analysis/, accessed on 10 May 2023. Significance vs. cut-off values between the lower and upper quartiles of expression are presented, with the red circle indicating the best cut-off.

**Figure 4 ncrna-09-00054-f004:**
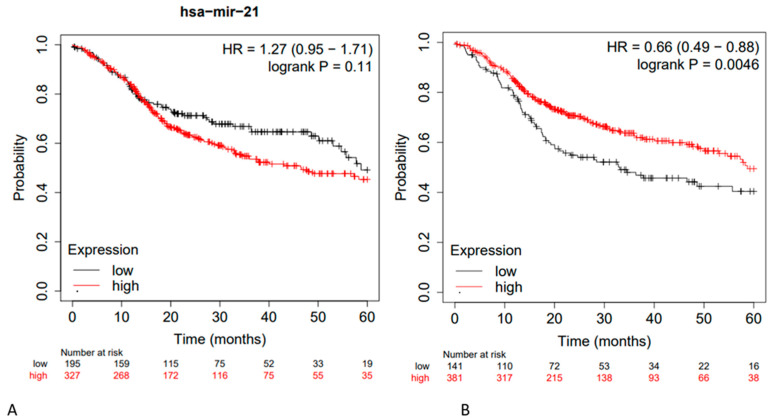
(**A**) Kaplan–Meier curve based on miR-21 expression levels for overall survival (OS) of patients with HNSCC (TCGA cohort) FDR: 100%; (**B**) Kaplan–Meier curve based on hsa-miR-153 (-), hsa-miR-200c (-), hsa-miR-363 (-), hsa-miR-17 (-), hsa-miR-205 (-), hsa-Let-7d (-), hsa-Let -7g (-), hsa-miR-34a (-), hsa-miR-375 (-), hsa-miR-491 (-), hsa-miR-218 (-), and hsa-miR-125b (-); OS HR = 0.66 95% CI: [0.46–0.88]; FDR: over 50%; Kaplan–Meier curves created from public database and Kaplan–Meier plotter web application (http://kmplot.com/analysis/, accessed on 10 May 2023).

**Table 1 ncrna-09-00054-t001:** Main data sources extracted from systematic reviews: National Heart, Lung, and Blood Institute (NHLBI); The Reporting Recommendations for Tumor Marker Prognostic Studies (REMARK); Quality Assessment of Diagnostic Accuracy Studies-2 (QUADAS-2); Meta-Analysis of Observational Studies in Epidemiology (MOOSE); The Newcastle–Ottawa Scale (NOS); tongue squamous cell carcinoma (TSCC); oral squamous cell carcinoma (OSCC); oropharyngeal squamous cell carcinoma (OPSCC); CI (confidence interval); The Cancer Genome Atlas (TCGA);\ data not reported; overall survival (OS); disease-free survival (DFS); recurrence-free survival (RFS); cancer-specific survival (CSS); progression-free survival (PFS); relative risk (RR).

First Author, Data	Country	miR	Studies and Reports Included	Patients/Sample Total Included	HR and RR Data Extract	Risk of Bias
Dioguardi et al., 2023 [9]	Italy	miR-195	3	81 TSCC, 304 LSCC	OS, RR = 0.36 95% CI: [0.25, 0.51];	REMARK
Dioguardi et al., 2022 [13]	Italy	miR-197	1 + TCGA	68 OSCC	OS, HR = 1.01, 95% CI: [1.00, 1.02];	REMARK
Dioguardi et al., 2022 [14]	Italy	miR-196a, miR-196b	5	417 HNSCC (105 TSCC, 3 OPSCC, 116 OSCC and others, 192 LSCC)	OS, HR = 1.67, 95% CI: [1.16, 2.49]; DFS, HR= 1.39, 95% CI: [0.33 5.52];	REMARK
Dioguardi et al., 2022 [8]	Italy	miR-155	8	709 HNSCC (120 LSCC)	OS, HR = 1.40, 95% CI: [1.13, 1.75]; DFS, HR = 1.36, 95% CI: [0.65 2.83]; PFS, HR = 1.09, 95% CI: [0.53 5.15]	REMARK
Dioguardi et al., 2022 [6]	Italy	miR-31	4	240 HNSCC	OS, HR = 1.58, 95% CI: [1.21, 2.06]	REMARK
Dioguardi et al., 2022 [7]	Italy	miR-21	8	351 OSCC	OS, HR = 1.29, 95% CI: [1.16, 1.4]; DFS, HR = 2, 95% CI: [1.35, 2.95]; CSS, HR = 1.19, 95% CI: [0.72, 1.97]; RFS, HR = 1.41, 95% CI: [0.48–4.15]	REMARK
Irimie-Aghiorghiese et al., 2019 [15]	Romania	miR-21	7	757 HNSCC	OS, HR = 1.719, 95% CI: [1.402, 2.109]	\
Lubov et al., 2017 [16]	Brazil, Canada	miR-21	4	456 HNSCC	OS, HR= 1.81, 95% CI: [0.66, 2.95]	QUADAS-2
Xie and Wu, 2017 [17]	China	miR-21	9	777 Oral Cancer	OS, HR = 1.71, 95% CI: [1.20, 2.44]; CSS, HR = 2.63, 95% CI: [1.25, 5.51]; RFS, HR = 2.04, 95% CI: [1.09, 3.80]; DFS, HR= 2.70, 95% CI: [1.08, 6.76]	MOOSE
Wang et al., 2019 [18]	China	miR-375	13 (5 HNSCC)	1340 patients (294 HNSCC)	HNSCC; OS, HR= 1.59, 95% CI [1.16, 2.18]	NOS, MOOSE
Li et al., 2019 [19]	China	miR-146ª	10 cancers (1 HNSCC + TGCA)	\	HNSCC; OS, HR = 0.734, 95% CI: [0.572, 0.941]	\
Qiu et al., 2021 [20]	China	miR-205, miR-429, miR-21, miR-331, miR-200a-3p, miR-19a, miR-21-5p, miR-151a, miR-17, miR-18b, miR-324, miR-96, miR-29c, miR-200b, miR-375, miRNA-204, miR−200c, miR-130a, miR-15b	10	1093 HNSCC	RFS, HR= 2.51, 95% CI: [2.13, 2.96]	NOS, QUADAS-2
Huang et al., 2021 [21]	China	miR-203, miR-195, miR-29c, miR-300, miR-146, miR-155, miR-200b, miR-16-2, miR-10a, miR-100, miR-101, miR-34c, miR-125, miR-149, miR-145, miR-181a, let-7a, miR-21, miR-494, miR-720, miR-675, miR-137, miR-31, miR-9, miR-19a, miR-424, miR-23a, miR-196b.	36	3020 LSCC	OS, HR = 1.10 95% CI: [1.00,1.20] downregolator; OS, HR = 1.13, 95% CI: [1.06–1.20] upregolator; DFS, HR = 2.57, 95% CI: [1.56–4.23].	NHLBI
Jamali et al., 2015 [22]	Iran	miR-34a, miR-375, mir-155, let-7g, miR-210, miR-20a, miR-126, miR-21, miR-205, miR-203, miR-19a, miR-134a, miR-200c, let-7b, miR-153, miR-18a, miR-17a, miR-451, miR-193b.	21	HNSCC	miR-21 OS, HR = 1.57, 95% CI [1.22–2.02];	MOOSE
Troiano et al. [23]	Italy	miR-21, miR-455-5p, miiR-155-5p, miR-372, miR-373, miR29b, miR-1246, miR-196a, miR-181. miR-204, miR-101, miR-32, miR-20a, miR-16, miR-17, miR-125b.	15	1200 OSCC	OS, HR = 2.65 95% CI: [2.07, 3.39]; DFS, HR 1.95 95% CI: [1.28, 2.98].	NOS

**Table 2 ncrna-09-00054-t002:** Data extracted from the 76 studies included, providing information regarding the type of tumor, the location of the tumor, the number of patients with data concerning the average age, the average or maximum follow-up, gender, and the common risk factors in the patients are reported to be smoking, alcohol, and HPV positivity; TNM (T: tumor size; N: regional lymph nodes; M: distant metastasis); pTNM, pathological TNM staging; cTNM, clinical TNM staging; N/A, not available; Ma (male); Fe (female); R (range); y (years); smoking (Sm); alcohol (Alc); SEM (standard error mean); PS (prospective study); RT (retrospective study); HPSCC (hypopharyngeal squamous cell carcinoma); OTSCC (oral tongue squamous cell carcinoma); BOTSCC (base of tongue squamous cell carcinoma); NPC (nasopharyngeal carcinoma). Data are not reported in a clear and explicit manner;\ data not present.

First Author, Date	Country	Study Design	Tumor Type/Tumor Site	miR	Follow-Up Months	Patient (Ma, Fe)	Age (Years)	Smoking	Alcohol	HPV	Staging
Jung et al., 2012 [24]	USA	RT	17 OSCC (Tongue base 6, Tongue anterior 3, Tongue border 2, Tongue ventral, Mouth 1, Oropharynx 1, Tongue unspecified 2)	miR-7, miR-21, miR-424	180	17 Ma	34–81	\	\	HPV +10, −7	pTNM stage I 1, II 5, IV 8, 3 N\A
Kawakita et al., 2014 [25]	Japan	RT	79 OTSCC	miR-21	60	44 Ma, 35 Fe	≥67 y 47, <67 y 32	\	\	\	T1 + T2 47, T3 + T4 32
Hedbäck et al., 2014 [26]	Denmark	RT	86 OSCC (Tongue 21, Mouth floor 65)	miR-21	60	63 Ma, 23 Fe	\	Sm yes 86	\	\	?
Yu et al., 2017 [27]	Taiwan	RT	100 OSCC (Buccal 37, Tongue 35, Mouth floor 12, Others 16)	miR-21	100	92 Ma, 8 Fe	55.3, ≤55 y 56, >55 y 44	\	\	\	Stage I + II 23, III + IV 77
Supic et al., 2018 [28]	Serbia	RT	60 OTSCC	miR-183, miR-21	80	47 Ma, 13 Fe	58, 43–82, <58 y 28, ≥58 y 32	Sm Never/former 18, current 42;	Alc low 39, Alc high 21	\	Stage II 15, III 45
Jakob et al., 2019 [29]	Germany	RT	36 OSCC (Mouth floor 6, Tongue 25, Palate 5)	miR-21, miR-29, miR-31, miR-99a, miR-99b, miR-100, miR-143, miR-155	58	27 Ma, 9 Fe	59, 23–84	Sm yes 28	Alc Yes 21	\	Stage I + II 10, III + IV 26
Li et al., 2013 [30]	China	RT	63 OSCC (Tongue)	miR-21	150	63 Ma	54, 35–72	\	\	\	\
Zheng et al., 2016 [31]	China	RT	84 Tongue cancer (72 OTSCC)	miR-21	90	\	\	\	\	\	\
Li et al., 2009 [32]	China	RT	103 OTSCC	miR-21	70	56 Ma, 47 Fe	<50 y 47, ≥50 y 56	\	\	\	Clinical Stage I + II 60, III + IV 43
Ganci et al., 2016 [33]	Italy	RT	92 OSCC	miR-130b, miR-141, miR-21, miR-96	60	57 Ma, 35 Fe	<64 y 48, >64 y 44	Sm never 22, Sm or ex 53	Alc no 31, Alc yes 43	HPV +1	T1 + T2 50, T3 + T4 42
Wang et al., 2018 [34]	China	RT	118 HNSCC	miR-31	60	65 Ma, 53 Fe	<56 y 51, ≥56 y 67	Sm yes 76, Sm no 42	Alc no 46, Alc yes 71	\	TNM stage I + II 33, III + IV 85
Qiang et al., 2019 [35]	China	RT	56 HNSCC (21 Hypopharynx, 25 Larynx)	miR-31	60	32 Ma, 24 Fe	≥60 y 29, <60 y 27	\	\	\	Clinical stage T1 + T2 21, T3 + T4 35
Tu et al., 2021 [36]	Taiwan	RT	40 OSCC	miR-31	160	36 Ma 4 Fe	57.53 ± 1.58 y	Sm yes 30	\	\	Stage I + III 12, IV 28
Hess et al., 2017 [37]	Germany	RT	149 HNSCC (Oropharynx 78, Hypopharynx 71)	miR-155, miR-200b, miR-146a	61	123 Ma, 26 Fe	57, 38–71	Ex Sm+Sm no 54, Sm yes 92	\	HPV-16 +12	\
Zhao et al., 2018 [38]	China	RT	120 LSCC (Glottic 74, Supraglottic 46)	miR-155	79	107 Ma, 13 Fe	≥60 y 63, <60 y 57	\	\	\	T1 + T2 67, T3 + T4 53
Baba et al., 2016 [39]	Japan	RT	73 OSCC	miR-155	50	49 Ma, 24 Fe	<60 y 18, ≧60 y 55	\	\	\	pTNM stage I + II 29, III + IV 44
Shi et al., 2015 [40]	China	RT	30 OSCC	miR-155	50	19 Ma, 11 Fe	40–75, 56.4 ± 8.6	Sm yes 16, never Sm 14	Alc no 16, Alc yes 14	\	stage I + II 8, III + IV 22
Kim et al., 2018 [41]	Korea	RT	68 OSCC (Oral Tongue 39, Buccal 13, Mouth floor 8, Retromolar trigone 7, Upper alveolar ridge 1)	miR-155	80	45 Ma, 23 Fe	57.7, 23–84	\	\	\	pTNM stage I + II 35, III + IV 33
Bersani et al., 2018 [42]	Sweden	RT	168 OTSCC/BOTSCC	miR-155, miR-185, miR-193b	34	126 Ma, 42 Fe	61	\	\	HPV +110	Tumor stage I + II 17, III + IV 155
Wu et al., 2020 [43]	China	RT	62 OSCC	miR-155	60	42 Ma, 20 Fe	≤50 y 39, >50 y 23	\	\	\	TNM stage I + II 46, III + IV 16
Shuang et al., 2017 [44]	China	PS	122 LSCC (Glottis 61, Supraglottis 42, Subglottis 19)	miR-195	60	80 Ma, 42 Fe	≤60 y 69, >60 y 53	Sm yes 99, Sm no 23	\	\	Clinical stage I + II 23, III + IV 99
Ding and Qi, 2019 [45]	China	PS	182 LSCC (Supraglottic 50, Glottic 95, Subglottic 37)	miR-195	60	120 Ma, 62 Fe	<60 y 80, ≥60 y 102	\	\	\	Clinical stage I + II 130, III + IV 52
Jia et al., 2013 [46]	China	PS	81 OTSCC	miR-195	48	45 Ma, 36 Fe	<60 y 45, ≥60 y 36	\	\	\	Clinical stage I + II 48, III + IV 33
Qin et al., 2019 [47]	China	PS	80 OSCC (Tongue 30, Gingival 24, Cheek 13 Floor of Mouth 10, Oropharynx 3)	miR-196a	80	43 Ma, 37 Fe	≥60 y 39, <60 y 41	Sm yes 30, Sm no 50	Alc no 56, Alc yes 24	\	TNM stage I + II 33, III + IV 47
Liu et al., 2013 [48]	Taiwan	PS	95 OSCC (Buccal 34, Tongue 25, Others 36)	miR-196a, miR-196a2	85	90 Ma, 5 Fe	53.6	\	\	\	Clinical stage I + III 26, IV 69
Maruyama et al., 2018 [49]	Japan	PS	50 OSCC (OTSCC 50)	miR-196a, miR-10a, miR-10b, miR-196b	6o	24 Ma, 26 Fe	<60 y 21, ≥60 y 29	Sm yes 19, Sm no 31	Alc no 25, Alc yes 22	\	Clinical stage I 32, II 18
Zhao et al., 2018 [50]	China	PS	113 LSCC (Glottic 70, Supraglottic 43)	miR-196b	97	96 Ma, 17 Fe	<60 y 42, ≥60 y 71	\	\	\	Tumor stage II 47, III + IV 66
Luo et al., 2019 [51]	China	PS	79 LSCC	miR-196b	60	66 Ma, 13 Fe	60.58	Sm yes 52, ex Sm 21, Sm no 6	exAlc 17, Alc no 4, Alc yes 58	\	TNM stage I + II 23, III + IV 56
Ahn et al., 2017 [52]	Korea	RT	68 OSCC	miR-197	44.3	45 Ma, 23 Fe	57.7, 23–84	\	\	\	pTNM I + II 35, III + IV 33
Hudcova et al., 2016 [53]	Czech Republic	RT	42 OSCC (34 patients included in the analysis)	miR-29c, miR-200b, miR-375	48	42 Ma	63, 47–87	\	\	\	Tumor stage T1 + T2 18, T3 + T4 22
Kang et al., 2021 [54]	China	RT	80 OSCC	miR-198	60	?	?	?	?	\	?
Bonnin et al., 2016 [55]	France	RT	75 Oropharynx (Base of tongue 24, Soft palate 11, Tonsil 22, Pharyngeal wall 4, Vallecula 9, Other 5)	miR-422a	50–120?	61 Ma, 14 Fe	54, 39–82	Alc yes + Sm yes 56	Alc yes + Sm yes 56	HPV +13	Staging III 13, S IV 62
Ganci et al., 2013 [56]	Italia	RT	121 HNSCC (Oral cavity 73, Larynx 29, Hypopharynx 9, Oropharynx 10)	miR-205, miR-429, miR-21, miR-331, miR-200a, miR-19a, miR-21, miR-151a, miR-17, miR-18b, miR-324, miR-96, miR-139, miR-21-5p, miR-17-3p	73	89 Ma, 32 Fe	<62 y 60, >62 y 60	Sm no 27, Sm yes or ex 94, Unknown 1	Alc yes or ex 70, Alc no 50, Unknown 1	HPV +114, −5, Unknown 2	pTNM T1 + T2 56, T3 + T4 65
Harris et al., 2012 [57]	USA	PS	123 HNSCC (OSCC 43, OPSCC 37, LSCC 43)	miR-375	60	85 Ma, 38 Fe	<58 y 45, ≥67 y 40, 59–66, 38	Sm never 18, ex Sm 57, Sm yes 48	Alc no 89, Alc yes 34	HPV +31, −74	TNM stage I + II 24, III + VI 99
Ahmad et al., 2019 [58]	Czech Republic	RT	94 patients, 43 cancers, (Oral cavity 8, Hipo-pharynx 13, Larynx 30, Oropharynx 43)	miR-15b	60	80 Ma, 14 Fe	58	\	\	\	TNM stage I + II 22, III + VI 72
Rajthala et al., 2021 [59]	Norway	RT	160 OSCC (Tongue 71, Gingiva 42, Buccal 20, Floor of mouth 18)	miR-204	103	102 Ma, 58 Fe	65.25, 27–93	Sm no 49, Sm yes 75	Alc low normal 51, Alc moderate-Hight 35	HPV −160	Stage Stage Ⅰ 43, stage Ⅱ 37, stage Ⅲ 25, Stage Ⅳ 57
Song et al., 2020 [60]	Japan	RT	204 OSCC (77 Tongue)	miR-200c	40	146 Ma, 58 Fe	<60 y 82, ≥60 y 122	Sm no 71, Sm yes 133	Alc no 36, Alc yes 168	\	TNM stage I + II 113, III + IV 91
Zhao et al., 2018 [61]	China	PS	132 LSCC (Glottic 76, Supraglottic 56)	miR-145	70	114 Ma, 18 Fe	<60 y 48, ≥60 84	\	\	\	T stage T2 51, T3 + T4 81
Li et al., 2013 [62]	China	RT	80 LSCC	miR-101	60	56 Ma, 24 Fe	≥60 y 48, <60 y 32	Sm no 60, Sm yes 20	\	\	Clinical stage I + II 38, III + IV 42
de Jong et al., 2015 [63]	Finland	RT	34 LSCC (supraglottic 18, glottic 16)	miR-452, miR-141, miR-203	60	20 Ma, 14 Fe	\	\	\	\	T stage 2–3 34
Fang et al., 2019 [64]	China	RT	66 LSCC (Supraglottic 19, Glottic 45, Subglottic 2)	miR-29c	110	62 Ma, 4 Fe	≤60 y 26, >60 y 40	\	Alc no 45, Alc yes 21	\	TNM stage I 7, II 13, III 14, IV 32
He et al., 2017 [65]	China	RT	133 LSCC	miR-300	60	87 Ma, 46 Fe	61.33 ± 7.86 y	\	\	\	TNM stage I + II 65, III + IV 68
Re et al., 2015 [66]	Italy	RT	99 LSCC (Supraglottic 19, Transglottic 66, Subglottic 5)	miR-34c	120	87 Ma, 3 Fe	66.51 ± 8.02 y	\	\	\	\
Xu et al., 2016 [67]	China	RT	97 LSCC (Glottic 31, supraglottic 19)	miR-149	80	73 Ma, 24 Fe	<60 y 46, ≥60 y 51, 70–35, 63.8	Sm (cigarette/day) 1–20 25, ≥20 36	Alc grams of <50 45, ≥50 52	\	Stages I + II 59, III + IV 38
Tian et al., 2014 [68]	China	RT	56 LSCC (Supraglottic 26, Glottic 30)	miR-203	60	40 Ma, 16 Fe	≥59 y 32, <59 y 24	\	\	\	Clinical stage I + II 24, III + IV 32
Zhao et al., 2018 [69]	China	RT	127 LSCC (Glottic 77, Supraglottic 50)	miR-181a	69	114 Ma, 13 Fe	≥60 y 79, <60 y 48	Sm no 12, Sm yes 115,	Alc no 93, Alc yes 34	\	T stage T2 53, T3 + T4 74
Guan et al., 2016 [70]	China	RT	65 HNSCC (Larynx 46, Hypo-others 16)	miR-675	72	48 Ma, 14 Fe	63.8, >64 y 33, ≥64 y 29	\	\	\	T stage T1 + T2 18, T3 + T4 44
Avissar et al., 2009 [71]	USA	RT	169 HNSCC (Oral 83, Pharyngeal 31, Laryngeal 19)	miR-375, miR-21	60	91 Ma 42 Fe	61.5 ± 11.9 y	Pack-years sm, No (0) 22, <36.75 43, ≥36.75 68	Drinks per week, No (0) 13, <18 50, ≥18 70	HPV\16, + 16	Stage I + I 46, III + IV 118
Wu et al., 2014 [72]	China	PS	103 LSCC (Supraglottic 66, Glottic 37)	miR-9	60	54 Ma, 49 Fe	<60 y 41, ≥60 y 62	\	\	\	TNM stage I + II 43, III + IV 60
Wu, Zhang et al., 2014 [73]	China	PS	83 LSCC	miR-19a	80	57 Ma, 26 Fe	≥56 y 42, <56 y 41	\	\	\	T stage T1 + T2 52, T3–T4 31
Zhang et al., 2015 [74]	China	RT	52 LSCC	miR-23a	60	45 Ma, 7 Fe	<60 y 22, ≥60 y 30	Sm no 7, Sm yes 45	Alc no 15, Alc yes 37	\	Clinical stage I 6, II 12, III 31, IV 3
Hu et al., 2015 [75]	China	RT	46 LSCC (Glottic 33, Supraglottic 11, Subglottic 2)	miR-21, miR-375	60	42 Ma, 4 Fe	59.2 ± 7.84 y	Sm no 12, Sm yes 31	Alc no 22, Alc yes 19	\	TNM stage I + II 31 III + IV 15
Re et al., 2017 [76]	Italy	RT	43 LSCC (Supraglottic 8, Transglottic 33, Subglottic 2)	miR-21, let-7a, miR-34c	55.7	42 Ma, 1 Fe	66.51 ± 8.02 y	\	\	\	TNM stage III 31 IV 12
Shen et al., 2012 [77]	China	RT	69 LSCC	miR-34a	40	\	<60 y 33, ≥60 y 36	\	\	\	TNM stage I + II 42, III + IV 27
Maia et al., 2017 [78]	Brazil, Singapore	RT	34 LSCC (Supraglottic 7, Glottic 27)	miR-296	40	30 Ma, 4 Fe	≤60 y 16, >60 y 18	Sm yes 31, Sm no 3	\	\	T stage I 16 II 18
Ogawa et al., 2012 [79]	Japan	RT	24 HNSCC (24 Sinonasal Squamous Cell Carcinomas)	miR-34a	53	16 Ma, 8 Fe	>60 y 14, <60 y 10	\	\	\	T stage T2 1, T3 10, T4a 13
Pantazis et al., 2020 [80]	Greece	RT	105 LSCC	miR-20b	84	60 Ma, 45 Fe	62, 36–87	\	\	\	TNM stage I 15, II 16, III 38, IV 36
Childs et al., 2009 [81]	USA	RT	94 HNSCC (Oral cavity 31, Oropharynx 32, Hypopharynx, 9 Larynx 32)	Let-7, miR-205, miR-21	60	71 Ma, 33 Fe	<60 y 41, <60 y 63	Sm current 46, Sm former 39, Sm never 17	\	HPV\16 −59, +37	Tumor stage I + II 24, III IV 80
Ko et al., 2014 [82]	Korea	RT	167 HNSCC (Oropharynx 88, Oral cavity 79)	miR-21	72	136 Ma, 31 Fe	56, 25–90	Sm no 57, Sm yes 109	\	HPV −131, +31	Stage I 26, II 35, III 20, IVa 86
Arantes et al., 2017 [83]	Brazil	RT	71 HNSCC (Oropharynx 35, Larynx/Hypopharynx 38)	miR-21	60	68 Ma, 3 Fe	40–76	Sm yes 57	Alc yes 27	HPV +6	Clinical stage T2 + T3 46, T4 25
Chang et al., 2013 [84]	Taiwan	RT	98 OSCC (Buccal 43, tongue 29, Gingiva 21, Floor of the mouth 5)	miR-20a, miR-17	84	83 Ma, 15 Fe	<50 y 34, >50 y 64	Sm yes 81, Sm no 17	\	\	Clinical stage I + II 42, III + IV 56
Gee et al., 2010 [85]	UK	RT	46 HNSCC (Oral cavity 10, Oropharynx 21, Hypopharynx 9, Larynx 5, Paranasal sinus 1)	miR-210, miR-21, miR-10b	60	37 Ma, 9 Fe	63, 43–92	Sm never 6, ex Sm 12, Sm current 28	Alc no 10, never heavy 14, currently heavy 22	\	Stage I 2, II 2, III 6, IV 35
Jia et al., 2014 [86]	China	RT	76 TSCC	miR-26a	48	40 Ma, 36 Fe	<60 y 41, ≥60 y 35	\	\	\	Clinical stage I+II 45, III+IV 31
Liao et al., 2013 [87]	China	RT	106 OSCC (Tongue 18 Floor of mouth 4 Buccal 12, Hard palate 4, Upper or lower gingival 11)	miR-1246	60	30 Ma, 19 Fe	<60 y 21, ≥60 y 28	\	\	\	TNM Stage I + II 25, III + IV 24
Liu et al., 2013 [88]	China	RT	280 NPC	miR-451	96	206 Ma 74 Fe	≤45 y 136, >45 y 144	\	\	\	TNM stage I + II 91, III + IV 189
Liu, Shen et al., 2013 [89]	Taiwan	RT	96 HNSCC (Buccal 34, Tongue 26, Oral pharynx and Other 36)	miR-134	80	90 Ma, 6 Fe	53.5	\	\	\	Stage I + III 27, IV 69
Luo et al., 2014 [90]	China	PS	168 NPC	miR-18a	80	127 Me, Fe 41	≥50 y 99, <50 y 69	\	\	\	Clinical stage I + II 72, III + IV 96
Peng et al., 2014 [91]	Taiwan	RT	58 OSCC	miR-218, miR-125b, Let-7g	60	\	\	\	\	\	\
Sasahira et al., 2012 [92]	Japan	RT	118 OSCC (Tongue 64, others 54)	miR-126	60	68 Me, 50 Fe	67.4, 46–91	\	\	\	Clinical stage I + II 74, III + IV 44
Tu et al., 2015 [93]	Taiwan	RT	50 OSCC (Buccal 17, Tongue 24, others 9)	miR-372, miR-373	150	47 Ma, 3 Fe	52.6	\	\	\	Stage I + II 8, III + IV 42
Wu et al., 2014 [94]	Taiwan	RT	115 OSCC (Tongue 60, Buccal 43, Lip\gingiva\plate 12)	miR-218	96	65 Ma, 50 Fe	<55 y 66, ≥55 y 49	Sm no 50, Sm yes 65	Alc no 63, Alc yes 53	HPV16/18 −55, +60	Stage I + II 61, III + IV 54
Xu et al., 2013 [95]	China	RT	65 OSCC	miR-153, miR-200c	60	\	\	\	\	\	\
Zhang et al., 2017 [96]	China	RT	44 OSCC	miR-375	60	\	\	\	\	\	\
Jia et al., 2015 [97]	China	RT	105 TSCC	miR-375	50	49 Ma, 56 Fe	<60 y 65, ≥60 y 40	\	\	\	Clinical stage Ⅰ + Ⅱ 59, Ⅲ + Ⅳ 46
Hu et al., 2014 [98]	China		46 LSCC (Glottic 33, Supraglottic 11, Subglottic 2)	miR-375, miR-21	60	42 Ma, 4 Fe	<65 y 22, ≤65 y 24	Sm no 12, Sm yes 31	Alc no 22, Alc yes 19	\	Stage I + II 31 III + IV 15
Gu et al., 2018 [99]	China		56 TSCC	miR-22	60	33 Ma, 23 Fe	>50 y 23, ≤50 y 33	\	\	\	Clinical stage II + IIIa 46, IIIb + IV 10

**Table 3 ncrna-09-00054-t003:** The values of HR (95% confidence interval) and RR for the different prognostic indices of survival are shown in the table; overall survival (OS); disease-free survival (DFS); recurrence-free survival (RFS); cancer-specific survival (CSS); progression-free survival (PFS); relative risk (RR); high versus low expression (H-L); low versus high expression (L-H); infinite (inf).

First Author, Date	miR	OS	DFS	CSS	RFS	PFS	RR
Jung et al., 2012 [24]	miR-21	5.31 (1.39–20.38) H-L					
Kawakita et al., 2014 [25]	miR-21			1.19 (0.71–1.9) H-L			
Hedbäck et al., 2014 [26]	miR-21		2.70 (1.1–6.9) H-L				
Yu et al., 2017 [27]	miR-21		1.87 (1.21–2.87) H-L				
Supic et al., 2018 [28]	miR-21		2.002 (0.904–4.434) H-L				
	miR-183		5.666 (1.708–18.791) H-L	1.868 (0.924–3.776) H-L			
Jakob et al., 2019 [29]	miR-21	2.31 (0.62–8.58) H-L			0.18 (0.02–1.39) H-L	0.16 (0.02–1.22) H-L	
	miR-29b	2,7726,7353.03 (0-inf) H-L			4.09 (0.93–17.93) H-L	4 (0.92–17.45) H-L	
	miR-31	3.69 (1.07–12.79) H-L			1.82 (0.66–5.05) H-L	2.31 (0.94–5.69) H-L	
	miR-99a	0.31 (0.1–0.95) H-L			0.69 (0.29–1.64) H-L	0.64 (0.28–1.42) H-L	
	miR-99b	0.58 (0.17–1.94) H-L			0.22 (0.07–0.76) H-L	0.27 (0.09–0.79) H-L	
	miR-100	3.14 (0.66–39.98) H-L			2.49 (0.72–8.67) H-L	2.85 (0.83–9.74) H-L	
	miR-143	0.2 (0.04–0.92) H-L			0.56 (0.22–1.45) H-L	0.46 (0.18–1.17) H-L	
	miR-155	2.94 (0.93–9.29) H-L			2.04 (0.67–6.2) H-L	1.92 (0.7–5.22) H-L	
Li et al., 2013 [30]	miR-21	2.13 (1.11–4.10) H-L					
Zheng et al., 2016 [31]	miR-21	1.22 (1.09–1.36) H-L					
Li et al., 2009 [32]	miR-21	2.06 (1.21–3.51) H-L					
Ganci et al., 2016 [33]	miR-21				4.2 (1.1–15.98) H-L		
	miR-130b				2.9 (0.8–11) H-L		
	miR-141				4 (1.26–13.9) H-L		
	miR-96				5.7 (1.52–21.3) H-L		
Wang et al., 2018 [34]	miR-31	3.31 (1.42–5.36) H-L	3.86 (1.53–6.05) H-L				
Qiang et al., 2019 [35]	miR-31	1.38 (1.02–1.87) H-L					
Tu et al., 2021 [36]	miR-31	1.68 (0.7747–3.6433) H-L					
Hess et al., 2017 [37]	miR-155	1.9 (1.0–3.7) H-L					
	miR-200b	1.4 (0.8–2.6) H-L					
	miR-146a	2.2 (1.2–4.3) H-L					
Zhao et al., 2018 [38]	miR-155	1.476 (0.983–1.916) H-L					
Baba et al., 2016 [39]	miR-155	5.156 H-L	1.3300 H-L				
Shi et al., 2015 [40]	miR-155	1.748 (0.508–6.015) H-L					
Kim et al., 2018 [41]	miR-155		1.6300 *p* = 0.7592 H-L				
Bersani et al., 2018 [42]	miR-155					0.5760 *p* = 0.30 H-L	
Wu et al., 2020 [43]	miR-155	1.6600 *p* = 0.6780 H-L	1.4900 *p* = 0.7861 H-L				
Shuang et al., 2017 [44]	miR-195						RR 0.358 (0.134–0.959)
Ding and Qi, 2019 [45]	miR-195						RR 0.3616 (0.2409–0.5428)
Jia et al., 2013 [46]	miR-195						RR 0.322 (0.120–0.865
Qin et al., 2019 [47]	miR-196a	2.175 (1.455–4.034) H-L					
Liu et al., 2013 [48]	miR-196a, miR-196a2		2.57(1.20–5.48) H-L				
Maruyama et al., 2018 [49]	miR-196a	0.91 (0.12–7.19) H-L	0.6 (0.18–2.06) H-L				
Zhao et al., 2018 [50]	miR-196b	1.577 (0.989–2.516) H-L					
Luo et al., 2019 [51]	miR-196b	1.80 (0.38–8.51) H-L					
Ahn et al., 2017 [52]	miR-197		1.01 (1.00–1.02)?				
Hudcova et al., 2016 [53]	miR-200b	1.00 (0.42–2.38) H-L	1.25 (0.51–3.08) H-L		0.91 (0.14–5.23) H-L		
	miR-375	1.32 (0.76–2.27) H-L	1.45 (0.74- 2.81) H-L		1.77 (0.67–468) H-L		
	miR-29c	0.89 (0.47–1.70) H-L	0.80 (0.37–1.75) H-L		0.31 (0.10–0.91) H-L		
Kang et al., 2021 [54]	miR-198	3.996 (1.345–5.885) L-H	3.609 (1.123–5.334) L-H				
Bonnin et al., 2016[55]	miR-422a				1.99 (1.07–3.7) L-H		
Ganci et al., 2013 [56]	miR-205				4.98 (1.67–14.9) H-L		
	miR-429				4.45 (1.59–12.45) H-L		
	miR-21-3p			2.17 (0.98–4.83) H-L	3.12 (1.28–7.6) H-L		
	miR-331				3.45 (1.24–9.64) H-L		
	miR-200a				3.1 (1.18–7.9) H-L		
	miR-19a				2.86 (1.1–7.7) H-L		
	miR-21-5p			2.41 (1.1–5.53) H-L	2.77 (1.04–7.38) H-L		
	miR-151a				3 (1–8.97) H-L		
	miR-17			2.1 (0.91–4.71) H-L	2.82 (0.98–8.14) H-L		
	miR-18b				2.54 (0.97–6.69) H-L		
	miR-324				2.62 (0.85–8) H-L		
	miR-96				2.19 (0.87–5.53) H-L		
	miR-139			0.33 (0.12–0.87) H-L			
Harris et al., 2012 [57]	miR-375	12.8 (3.4–48.6) L-H					
Ahmad et al., 2019 [58]	miR-15b				0.246 (0.053–0.787) H-L		
Rajthala et al., 2021 [59]	miR-204	0.668 (0.45–1.00) H-L			0.56 (0.33–0.96) H-L		
Song et al., 2020 [60]	miR-200c	1.669 (1.03–2.703) L-H			1.705 (1.136–2.56) L-H		
Zhao et al., 2018 [61]	miR-145	0.662 (0.298–1.004) H-L					
Li et al., 2013 [62]	miR-101	1.13 (0.17–7.50) L-H					
de Jong et al., 2015 [63]	miR-452				0.5 *p* = 0.1 H-L		
	miR-141				0.7 *p* = 0.4 H-L		
	miR-203				0.6 *p* = 0.4 H-L		
Fang et al., 2019 [64]	miR-29c	0.350 (0.129–0.949) H-L					
He et al., 2o17 [65]	miR-300	1.89 (0.66–2.33) L-H					
Re et al., 2015 [66]	miR-34c	3.623 (1.911–6.86) L-H	1.81 (1.02–3.25) L-H				
Xu et al., 2016 [67]	miR-149	1.57 (1.02–2.40) L-H					
Tian et al., 2014 [68]	miR-203	*p* = 0.002 L-H					
Zhao et al., 2018 [69]	miR-181a	0.559 (0.211–1.106) H-L					
Guan et al., 2016 [70]	miR-675	2.52 (1.75–8.45) H-L	3.26 (0.94–10.71) H-L				
Avissar et al., 2009 [71]	miR-21	1.68 (1.04–2.77) H-L					
Wu et al., 2014 [72]	miR-9	3.18 (2.19–11.91) H-L					
Wu, Zhang et al., 2014 [73]	miR-19	2.260 *p* = 0.034 H-L					
Zhang et al., 2015 [57]	miR-23a	6.712 (2.076–21.700) H-L					
Hu et al., 2015 [75]	Expression ratio of miRNA-21/miRNA-375	*p* = 0.032					
Re et al., 2017 [76]	miR-34c-5p	7.32 (2.33–23.00) L-H	7.830 (2.225–27.552) L-H				
Shen et al., 2012 [77]	miR-34a		4.02 (1.67–9.69) L-H				
Maia et al., 2017 [78]	miR-296		8.6 (1.7–42.2) H-L				
Ogawa et al., 2012 [79]	miR-34a		0.005 (0.00–0.29) L-H	0 L-H?			
Pantazis et al., 2020 [80]	miR-20b	11.62 (2.64–46.62) H-L	4.23 (1.75–22.52) H-L				
Childs et al., 2009 [81]	miR-205	2.51 p 0.025 L-H					
	miR-21	1.00 p 0.995 H-L					
	Le7d	1.73 p 0.166 L-H					
Ko et al., 2014 [82]	miR-21			2.972 (1.340–6.590) L-H?	1.659 (0.824–3.343) L-H?		
Arantes et al., 2017 [83]	miR-21	2.05 (1.05–4.02) H-L					
Chang et al., 2013 [84]	miR-17	2.47 (1.37–4.44) L-H					
	miR-20a	3.44 (1.45–8.15) L-H					
Gee et al., 2010 [85]	miR-210	6.88 (2.30–20.53) H-L					
Jia et al., 2014 [86]	miR-26a						RR 0.283 (0.118–0.682)
Liao et al., 2013 [87]	miR-1246	2.82 (1.07–7.43) H-L					
Liu et al., 2013 [88]	miR-451	2.00 (1.18–3.41) L-H	1.81 (1.16–2.83) L-H				
Liu, Shen et al., 2013 [89]	miR-134	2.17 (1.17–5.12) H-L					
Luo et al., 2014 [90]	miR-18a	0.4147 (0.2208–0.7791) L-H					
Peng et al., 2014 [91]	Let-7g		3.267 (1.164–9.174) L-H	3.289 (1.059–10.204) L-H			
Sasahira et al., 2012 [92]	miR-126		2.631 (0.9886–7.9851) L-H				
Tu et al., 2015 [93]	miR-372		2.57 (1.20–5.48) H-L				
	miR-373		2.62 (1.47–4.64) H-L				
Wu et al., 2014 [77]	miR-218	2.51 (1.32–4.77) L-H					
Xu et al., 2013 [95]	miR-153	2.295 (1.168–4.508) L-H					
	miR-200c	2.202 (1.110–4.371) L-H					
Zhang et al., 2017[96]	miR-375	1.61 (0.96–2.70) L-H					
Jia et al., 2015 [97]	miR-375	2.07 (1.02–4.20) L-H					RR 0.449 (0.207–0.978)
Hu et al., 2014 [98]	miR-375	1.88 (0.56–6.31) L-H					
	miR-21	1.1302 (0.34–3.757) H-L					
Gu et al., 2018 [99]	miR-22		*p* < 0.005 L-H				

**Table 4 ncrna-09-00054-t004:** Spearman’s correlations.

miR	miR-21	miR-155	miR-375
miR-21	1 (*p* < 1 × 10^−4^)		
miR-155	0.1122 (*p* = 0.0152)	1 (*p* < 1 × 10^−4^)	
miR-375	−0.4331 (*p* < 1 × 10^−4^)	−0.0107 (*p* = 0.8177)	1 (*p* < 1 × 10^−4^)

**Table 5 ncrna-09-00054-t005:** Pearson’s correlations among different microRNAs.

miR	miR-21	miR-155	miR-375
miR-21	1 (*p* < 1 × 10^−4^)		
miR-155	0.1426 (*p* = 0.002)	1 (*p* < 1 × 10^−4^)	
miR-375	−0.2241 (*p* < 1 × 10^−4^)	−0.0336 (*p* = 0.4684)	1 (*p* < 1 × 10^−4^)

**Table 6 ncrna-09-00054-t006:** Data on resistance to chemotherapy and radiotherapy in relation to altered expression of microRNAs.

First Autor, Data	miR	Tumor Type	Adjuvant Therapy	Administered Chemotherapy Drug	Main Results of the Study
Zheng et al., 2016 [31]	miR-21	OTSCC	chemotherapy	\	miR-21 enhances chemo-resistance in OTSCC
Hess et al., 2017 [37]	miR-155, miR-200b, miR-146a	HNSCC	radiotherapy/chemotherapy	5-fluorouracil/cisplatin, 5-fluorouracil/mitomycin C	MiR-146a was revealed as a prognostic marker for chemoradiation. MiR-155 and miR-146a were identified as markers for tumor-infiltrating lymphocytes.
Qin et al., 2019 [47]	miR-196a	HNSCC	chemotherapy	cisplatin	miR-196a may serve as a promising predictor of and potential therapeutic target for cisplatin resistance in HNC
Ahmad et al., 2019 [58]	miR-15b	HNSCC	radiotherapy	\	miR-15b-5p represents a potentially helpful biomarker for individualized treatment decisions concerning the management of HNSCC patients treated with intensity-modulated radiotherapy
de Jong et al., 2015 [63]	miR-203	HNSCC	radiotherapy	\	miR-203 causes intrinsic radioresistance of HNSCC, which could enable the identification and treatment modification of radioresistant tumors.
Maia et al., 2017 [78]	miR-296	LSCC	radiotherapy	\	miR-296-5p expression is associated with resistance to radiotherapy and tumor recurrence in early-stage LSCC
Ogawa et al., 2012 [79]	miR-34a	HNSCC	chemotherapy	cisplatin	miR-34a expression can be an independent prognostic biomarker in patients with sinonasal squamous cell carcinoma who are undergoing treatment with cisplatin
Zhang et al., 2017 [96]	miR-375	OSCC	radiotherapy	\	miRNA-375 inhibits growth and enhances radiosensitivity in OSCC
Gu et al., 2018 [99]	miR-22	TSCC	chemotherapy	cisplatin	strong correlation between miR-22 expression and chemosensitivity to cisplatin in TSCC patients

**Table 7 ncrna-09-00054-t007:** Risk of bias, ROBIS scale: ok (low); ? (unclear).

	Phase 1	Phase 2				Phase 3
First Author, Data	PICO	Study Eligibility Criteria	Identification and Selection of Studies	Data Collection and Study Appraisal	Synthesis and Findings	Risk of Bias in the Review
Dioguardi et al., 2023 [9]	ok	ok	ok	ok	?	ok
Dioguardi et al., 2022 [13]	ok	ok	ok	ok	ok	ok
Dioguardi et al., 2022 [14]	ok	ok	ok	ok	ok	ok
Dioguardi et al., 2022 [8]	ok	ok	ok	ok	ok	ok
Dioguardi et al., 2022 [6]	ok	ok	ok	ok	ok	ok
Dioguardi et al., 2022 [7]	ok	ok	ok	ok	ok	ok
Irimie-Aghiorghiese et al., 2019 [15]	ok	?	?	ok	ok	ok
Lubov et al., 2017 [16]	ok	?	?	?	ok	ok
Xie and Wu, 2017 [17]	ok	?	ok	ok	ok	ok
Wang et al., 2019 [18]	ok	?	ok	ok	ok	ok
Li et al., 2019 [19]	ok	?	?	?	ok	ok
Qiu et al., 2021 [20]	ok	ok	ok	ok	ok	ok
Huang et al., 2021 [21]	ok	?	?	ok	ok	ok
Jamali et al., 2015 [22]	ok	?	ok	ok	ok	ok
Troiano et al., 2018 [23]	ok	?	ok	ok	ok	ok

**Table 8 ncrna-09-00054-t008:** Assessment of the risk of bias; REMARK.

First Author, Data	Sample	Clinical Data	Marker Quantification	Prognostication	Statistics	Classical Prognostic Factors	Score
Jung et al., 2012 [24]	1	2	3	2	2	3	13
Kawakita et al., 2014 [25]	3	2	2	2	1	3	13
Hedbäck et al., 2014 [26]	3	2	2	3	3	2	15
Yu et al., 2017 [27]	3	2	3	3	3	2	16
Supic et al., 2018 [28]	2	3	3	3	3	3	17
Jakob et al., 2019 [29]	1	3	3	3	3	3	16
Li et al., 2013 [30]	2	2	3	2	2	3	14
Zheng et al., 2016 [31]	3	1	3	3	2	2	14
Li et al., 2009 [32]	3	3	3	3	3	3	18
Ganci et al., 2016 [33]	3	2	3	3	3	2	16
Wang et al., 2018 [34]	3	2	3	2	2	3	15
Qiang et al., 2019 [35]	2	3	3	2	2	2	14
Tu et al., 2021 [36]	1	3	3	2	2	2	13
Hess et al., 2017 [37]	3	2	3	2	2	2	14
Zhao et al., 2018 [38]	3	2	3	2	2	2	14
Baba et al., 2016 [39]	2	3	3	2	2	2	14
Shi et al., 2015 [40]	1	2	3	2	2	2	12
Kim et al., 2018 [41]	2	2	3	2	2	1	12
Bersani et al., 2018 [42]	3	3	3	2	2	1	14
Wu et al., 2020 [43]	2	2	3	3	3	2	15
Shuang et al., 2017 [44]	3	2	3	2	3	2	15
Ding and Qi, 2019 [45]	3	2	3	2	3	2	15
Jia et al., 2013 [46]	2	2	3	2	3	2	14
Qin et al., 2019 [47]	2	3	2	2	3	2	14
Liu et al., 2013 [48]	2	1	2	2	3	2	12
Maruyama et al., 2018 [49]	2	2	2	3	2	3	14
Zhao et al., 2018 [50]	3	1	2	2	3	2	13
Luo et al., 2019 [51]	2	3	3	2	2	2	14
Ahn et al., 2017 [52]	2	3	1	2	3	2	14
Hudcova et al., 2016 [53]	2	2	3	3	3	2	15
Kang et al., 2021 [54]	2	1	3	3	3	2	14
Bonnin et al., 2016 [55]	2	2	3	3	3	2	15
Ganci et al., 2013 [56]	3	3	3	3	3	2	17
Harris et al., 2012 [57]	3	3	2	3	3	2	16
Ahmad et al., 2019 [58]	2	2	3	3	3	2	15
Rajthala et al., 2021 [59]	3	3	3	3	3	2	17
Song et al., 2020 [60]	3	2	3	3	3	2	16
Zhao et al., 2018 [61]	3	2	3	2	3	2	15
Li et al., 2013 [62]	2	3	3	3	2	2	15
de Jong et al., 2015 [63]	1	1	3	2	3	3	13
Fang et al., 2019 [64]	2	3	3	3	3	2	16
He et al., 2017 [65]	3	2	2	3	3	2	15
Re et al., 2015 [66]	3	1	2	3	3	2	14
Xu et al., 2016 [67]	3	3	2	3	2	3	16
Tian et al., 2014 [68]	2	1	2	3	2	3	13
Zhao et al., 2018 [69]	3	3	2	3	3	2	16
Guan et al., 2016 [70]	2	2	2	3	3	3	15
Avissar et al., 2009 [71]	3	3	3	3	3	3	18
Wu et al., 2014 [72]	3	2	2	3	3	3	16
Wu, Zhang et al., 2014 [73]	2	3	2	3	3	3	16
Zhang et al., 2015 [74]	1	3	2	3	2	3	14
Hu et al., 2015 [75]	1	3	3	3	2	3	15
Re et al., 2017 [76]	1	2	3	3	2	3	14
Shen et al., 2012 [77]	2	1	2	3	2	3	13
Maia et al., 2017 [78]	1	3	2	3	2	2	13
Ogawa et al., 2012 [79]	1	1	2	3	2	3	12
Pantazis et al., 2020 [80]	3	2	2	3	3	3	16
Childs et al., 2009 [81]	2	3	3	3	3	3	17
Ko et al., 2014 [82]	3	3	2	3	3	3	17
Arantes et al., 2017 [83]	2	3	2	3	2	3	15
Chang et al., 2013 [84]	2	2	3	3	2	3	15
Gee et al., 2010 [85]	1	3	3	3	2	3	15
Jia et al., 2014 [86]	2	2	2	3	2	3	14
Liao et al., 2013 [87]	3	2	2	2	2	3	14
Liu et al., 2013 [88]	3	2	2	3	3	3	16
Liu, Shen et al., 2013 [89]	2	2	2	3	2	3	14
Luo et al., 2014 [90]	3	2	2	3	3	3	16
Peng et al., 2014 [91]	1	1	3	2	2	3	12
Sasahira et al., 2012 [92]	3	2	2	2	2	3	14
Tu et al., 2015 [93]	1	2	3	3	2	3	14
Wu et al., 2014 [94]	3	3	2	3	3	3	17
Xu et al., 2013 [95]	2	1	3	2	2	3	13
Zhang et al., 2017 [96]	1	1	2	2	2	3	11
Jia et al., 2015 [97]	3	2	2	2	2	3	14
Hu et al., 2014 [98]	1	3	3	2	3	3	15
Gu et al., 2018 [99]	2	2	2	2	3	3	14

## Data Availability

Not applicable.

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
