# Peer review of "The Potential microRNA Prognostic Signature in HNSCCs: A Systematic Review"

_ncrna, 2023, doi:10.3390/ncrna9050054_

Round 1

Reviewer 1 Report

This is a systemic review of articles and based on that authors reported three miRNAs- miR-21, miR-155 and miR-375 in head and neck cancer samples (HNSCC). Authors evaluated correlation among the three miRNAs where, miR-375 is negatively correlated with others. Also combined high and low-expression of the miRNAs were not associated with over-all survival (OS) as prognostic indication. Additionally miR-21 high vs. low expression showed no significant correlation with the OS. Authors' findings regarding the role of miRNAs in HNSCC prognosis are inconclusive.

In general, authors have searched a potential number of articles for identification of miRNAs. However, criteria for selecting only three miRNAs- miR-21, miR-155 and miR-375- among 64 miRNAs are not clear and strong. As research on non-coding RNAs in cancer are in developing stages, all the studies have equal importance in clinical perspective and disease management. Some of the miRNAs are already in clinical trial for diagnostic and prognostic studies of HNSCC.

The sample sizes acquired from the various researches are quite good enough, therefore there shouldn't be a problem with any inclusive results (lines 292-293).

As prognostic markers, authors have focused only on OS, ignoring disease free survival, metastasis, virus positivity, drug resistance, gene alterations etc.

The low and high expression cut-off was auto-generated and thus it is not clear.

The Abstract should be more specific to the article, especially methods, observation and outcomes.

Author Response

Replies to comments are in the attached word file

Best regards Mario Dioguardi

Reviewer 2 Report

Significance:

 A review by Mario Dioguardi et. al. “Potential microRNA Prognostic Signature in HNSCC: Systematic Review” is highlighting risk factors miRNAs as biomarkers for HNSCC, which is the 6th leading cancer worldwide.

Comments:

·        Plethora of miRNA expression datasets are available for control and cancer TCGA and other consortiums including the data for HNSCC cancer.  but it is not clear why the authors wish to do this analysis on individual studies over substantial databases.

·        Please includes the ways to bring identified biomarkers to the clinic for early diagnostic purposes.

Can be improved in the discussion and in the result part of the MS.

Author Response

(The authors gave the same response as above.)

Reviewer 3 Report

A main objection concerning  the reviewed study is a lack of positive conclusion coming from undertaken studies. Too much extent is connected with heterogeneity of head and neck cancer. It concerns almost everything including causative factors, biology, course of disease and treatment. So, how it could be expected to get a single clear biomarker of prediction?

Making a selection of papers useful for finding  of workable biomarker the author rely mostly on own publications (6 v. 8). Are other authors not clever enough to provide valuable data?

Altogether,  in my opinion the study is not worth to be published. Perhaps we (and the authors) should wait for some time to get more data to be analyzed.

Author Response

(The authors gave the same response as above.)

Round 2

Reviewer 1 Report

The manuscript has been properly revised. However, authors should double-check their writing for mistakes. For example: “..have an essential role in controlling the expression level of genes at the post- post-transcriptional level.”- Lines 22-23.

Additionally, the Abstract needs to end with a conclusion.

Based on the prospective functions exhibited by miRNAs in HNSCC—particularly miR-21, miR-155, and miR-375—a conclusion is required. 

Author Response

Reviewer 1

The manuscript has been properly revised. However, authors should double-check their writing for mistakes. For example: “..have an essential role in controlling the expression level of genes at the post- post-transcriptional level.”- Lines 22-23.

Additionally, the Abstract needs to end with a conclusion.

Based on the prospective functions exhibited by miRNAs in HNSCC—particularly miR-21, miR-155, and miR-375—a conclusion is required. 

ANSWER

Thank you for reviewing the manuscript and your suggestions have been very helpful in improving the manuscript.

All suggested corrections have been modified and the results with conclusions have been added to the abstract

Best regards Mario Dioguardi

Reviewer 2 Report

The abstract can be improved and can be more concise.

Please double-check the grammar and spelling errors throughout the MS 

Author Response

Reviewer 2

The abstract can be improved and can be more concise.

Please double-check the grammar and spelling errors throughout the MS

Answer

Thank you for reviewing the manuscript, your suggestions have been very helpful in improving the manuscript.

The abstract has been made more concise and the text has been revised to check for grammatical and spelling errors

Best Regards Mario Dioguardi